# Effect of Transcranial Direct Current Stimulation Augmented with Motor Imagery and Upper-Limb Functional Training for Upper-Limb Stroke Rehabilitation: A Prospective Randomized Controlled Trial

**DOI:** 10.3390/ijerph192215199

**Published:** 2022-11-17

**Authors:** Faizan Zaffar Kashoo, Raid Saleem Al-Baradie, Msaad Alzahrani, Ahmad Alanazi, Md Dilshad Manzar, Anchit Gugnani, Mohammad Sidiq, Mohammad Abu Shaphe, Mohamed Sherif Sirajudeen, Mehrunnisha Ahmad, Bader Althumayri, Abdullah Aljandal, Ahmed Almansour, Shady Abdullah Alshewaier, Aksh Chahal

**Affiliations:** 1Department of Physical Therapy and Health Rehabilitation, College of Applied Medical Sciences, Al Majmaah 11952, Saudi Arabia; 2Department of Medical Laboratory Sciences, College of Applied Medical Sciences, Majmaah University, Al Majmaah 11952, Saudi Arabia; 3Department of Nursing, College of Applied Medical Sciences, Al Majmaah 11952, Saudi Arabia; 4NIMS College of Physiotherapy and Occupational Therapy, NIMS University Jaipur, Jaipur 303121, Rajasthan, India; 5Department of Physical Therapy, College of Applied Medical Sciences, Jazan University, Jazan 82511, Saudi Arabia; 6Department of Physical Therapy, Security Forces Hospital, Riyadh 11564, Saudi Arabia; 7Department of Physical Therapy and Rehabilitation, Al Fayha Club, Al Majmmah 11952, Saudi Arabia; 8Maharishi Markandeshwar Institute of Physiotherapy and Rehabilitation, Maharishi Markandeshwar, Mullana 133207, Haryana, India

**Keywords:** motor imagery, chronic stroke, transcranial direct current stimulation, upper-limb rehabilitation, Fugl-Meyer’s scale, action research arm test

## Abstract

Background: Combining transcranial direct current stimulation (tDCS) with other therapies is reported to produce promising results in patients with stroke. The purpose of the study was to determine the effect of combining tDCS with motor imagery (MI) and upper-limb functional training for upper-limb rehabilitation among patients with chronic stroke. Methods: A single-center, prospective, randomized controlled trial was conducted among 64 patients with chronic stroke. The control group received sham tDCS with MI, while the experimental group received real tDCS with MI. Both groups performed five different upper-limb functional training exercises coupled with tDCS for 30 min, five times per week for two weeks. Fugl-Meyer’s scale (FMA) and the Action Research Arm Test (ARAT) were used to measure the outcome measures at baseline and after the completion of the 10th session. Results: Analysis of covariance showed significant improvements in the post-test mean scores for FMA (F (414.4) = 35.79, *p* < 0.001; η^2^ = 0.37) and ARAT (F (440.09) = 37.46, *p* < 0.001; η^2^ = 0.38) in the experimental group compared to the control group while controlling for baseline scores. Conclusions: Anodal tDCS stimulation over the affected primary motor cortex coupled with MI and upper-limb functional training reduces impairment and disability of the upper limbs among patients with chronic stroke.

## 1. Introduction

The second largest cause of mortality and disability worldwide is stroke [1]. Patients with stroke suffer from neuromuscular disabilities, including impairments in motor control [2]. The majority of patients with stroke suffer from hemiplegia (one-sided muscular paralysis). Stroke survivors with hemiplegia exhibit more upper-limb (UL) than lower-limb (LL) disability [3]. Investigations exploring patterns of stroke recovery have seen a steady increase in research focus in recent years. Based on the results of these studies, a significant proportion of the recovery occurs within the initial 30 days of a stroke, but improvement can occur for up to 12 months afterward [4]. The severity of the neurological deficits and early patterns of improvement are the two best predictors of recovery from impairments. Patients with early stroke who report early and immediate changes in their motor function have a much higher level of recovery than those who do not [5]. Patients are generally thought to experience less UL motor recovery than LL motor recovery; however, this clinical belief is typically based on disability assessments rather than tests of specific motor deficits of the UL and LL. Young patients with stroke experiencing severe motor impairment in the lower extremities may have functional gait (i.e., significant impairment but limited disability) [6]. Since UL function needs finer motor control than LL function, this might explain the common scenario of less variation between impairment and disability. A study reported that the pattern of recovery for the upper and lower limbs was similar after controlling the results for the severity of the stroke [7]. Therefore, it would be useful to develop a safe, easy, and clinically feasible treatment technique to reduce upper-limb disability. There are several promising treatment options available for upper-limb stroke rehabilitation, such as cross education [8], constraint-induced movement therapy [9], virtual reality therapy [10], functional electrical stimulation [11], robotic therapy [12], anodal transcranial direct current stimulation [13], and motor imagery [14].

Motor imagery is the cognitive rehearsal of the physical task without voluntary physical movement. Visualization of the physical activity is carried out after watching a videotape or listening to an audiotape. Motor imagery can be performed by imagining how the movement will look (visual motor imagery) or by imagining how the muscles will feel as they contract and move (kinesthetic motor imagery) [15]. A systematic review of four randomized controlled trials and one clinical trial reported a positive effect of mental rehearsal on arm function after a stroke [16], notwithstanding coupling MI with other treatments may be more effective than motor imagery alone [17].

One of those treatments that could be easily coupled with MI is transcranial direct current stimulation (tDCS). tDCS is a noninvasive brain-stimulation technique used to modulate specific areas of cortical activity. Studies have reported that tDCS offers a potential treatment for a number of neurological disorders [18]. Much research has evaluated how tDCS affects recovery from stroke, particularly motor function [19,20,21]. For example, one study using tDCS reported improvement in the forearm motor function of stroke patients, reporting a 10–30% improvement in their forearm motor function [22].

Research studies reported that pairing tDCS with the behavior to be modulated may offer away to achieve the desired effect [23]. As of now, the effects of transcranial direct current stimulation (tDCS) on stroke patients have been studied primarily without any specific behavioral therapy [24]. These trials have demonstrated relatively modest functional gains. Given that tDCS affects the majority of neuronal circuits in an untargeted manner, this strategy is likely unsatisfactory. Instead, MI paired with transcranial direct current stimulation (tDCS) may drive plastic processes toward a functional outcome. tDCS may have focally positive effects by activating neuronal networks in the injured hemisphere. Yet, the effects of tDCS are dependent on the brain’s baseline state at the time of application. Therefore, the purpose of this study was to examine the effect of tDCS to modulate the baseline state of the brain in combination with visual motor imagery and upper-limb functional training in patients with chronic stroke. We hypothesize that tDCS combined with MI and upper-limb motor training would significantly improve the upper-limb motor function following chronic stroke compared to MI and upper-limb motor training alone. 

## 2. Materials and Methods

### 2.1. Trial Design

This study was conducted from April 2017 to September 2022 as a prospective, randomized controlled trial at a single center with the equal allocation (1:1) to two treatment arms. This research was conducted at the University Hospital of the National Institute of Medical Sciences, Rajasthan in Jaipur, India. The trial was approved by the institutional review board in February 2017. In April 2017, the study protocol was registered with the US trial registry (clinical trial ID: NCT03122821).

### 2.2. Participants

The criteria for inclusion were: participants with chronic stroke (stroke in the past 6 months), aged between 18 and 80 years. Participants who could perform visual motor imagery were assessed with the 10-item Kinesthetic and Visual Imagery Questionnaire (KVIQ-10). The KVIQ-10 was reported to be suitable for patients with disabilities [25]. The criteria for exclusion were: cranial implants, pacemakers, or artificial implanted hearing aids; a history of convulsions; brain surgery or head injury; and speech disorder (aphasia), unilateral neglect, or impaired cognition (Mini-Mental State Examination score less than 24) [26]. The study procedures followed the guidelines of the Helsinki Declaration (Figure 1).

### 2.3. Randomization and Blinding

Using a computer-generated block of four numeric randomization codes, participants were randomly assigned to either the experimental group or the control group. These codes were generated by members of the study team who were not involved in participant assessment or recruitment. The participants were blinded to their group assignment, as were the physiotherapists who completed the clinical outcome evaluations. The physiotherapists involved in the assessment had more than 10 years of experience in the assessment and treatment of patients with stroke. 

### 2.4. Sample Size Calculation

As analysis of covariance (ANCOVA) was the primary method of analysis, the partial eta squared (η^2^ = 0.43) from a previous study was used to calculate the sample size [27]. The sample size was estimated using G*Power 3.1.7 with the parameter power of 0.80, critical F of 3.17, and alpha error probability of 0.05, resulting in a sample size of 56 participants. The optimal sample size was 64, assuming a dropout rate of 10–15%.

### 2.5. Outcome Variables

#### 2.5.1. Fugl-Meyer Assessment of Upper-Limb Motor Recovery following Stroke

The Fugl-Meyer Assessment of Motor Recovery After Stroke (FMA) [28] is a 66-point assessment tool used to evaluate impairment of the upper extremities based on a three-point ordinal scale. A zero score implies non-performance, one implies partial performance, and two implies complete performance. The FMA is reported to have excellent reliability (total = 0.98–0.99; subtests = 0.87–1.00) and validity [29,30,31,32,33,34]. It is commonly used in clinical trials conducted to assess changes in motor impairment following stroke [35,36,37].

#### 2.5.2. Action Research Arm Test (ARAT)

This test employs a four-point ordinal quantitative scale (a score of zero denotes no movement, a score of one denotes movement partially completed, two denotes movement performed abnormally, and three denotes movement performed normally), giving a total raw score of 56. This test has shown high inter-rater (0.98) and test-retest reliability (0.99) [38].

### 2.6. Protocol for Visual-Motor Imagery

The visual mental imagery (VMI) protocol was adapted from previous research conducted among stroke patients [39]. The participants were shown video and audio tapes of the activities to be performed on the hand rehabilitation table. The initial 15 min were used to prepare video and audio tapes for the activity while the participants relaxed in a quiet room. This was followed by asking participants to watch the sequence of movements performed on the video tape, then listen to the same activity on an audio tape. After running the video and audio tapes two to three times, participants were connected to the tDCS device (Figure 2). The mental imagery of the movement was performed while the patient concurrently received tDCS stimulation over the C3/C4 region of the cerebral cortex. This was followed by the actual performance of the activity. After five attempts at the activity, the participants repeated VMI. Participants repeated the activity as many times as possible with a rest period of less than a minute. Each participant was supervised during the activity at their normal functional pace until the full 30 min session was completed. There were five activities completed in the first week, followed by a repetition of these activities in the second week (Figure 3).

### 2.7. Protocol for Transcranial Direct Current Stimulation

A pair of saline-soaked surface sponge electrodes (35 cm^2^) and a battery-powered, constant-current stimulator were utilized to transmit direct current (The Brain Stimulator v3.0 Deluxe; sunny Southern California, Sacramento, CA, USA). To facilitate the primary motor cortex (M1), an anode electrode was positioned over C3/C4 (International 10/20 Electroencephalogram System), which corresponds approximately to the upper-limb motor cortex of the affected hemisphere (Figure 2). The cathode electrode was positioned on the contralateral supraorbital region. During the first 30 min of MI training, patients received 1.5 mA of tDCS for active stimulation [40]. tDCS was administered for a length of 30 min in compliance with existing safety restrictions [41]. The tDCS protocol (1.5 mA of anodal tDCS for 30 min) was chosen based on previous research [40]. In the sham circumstances, a current ran for 30 s at the onset of stimulation and then stopped [42].

### 2.8. Upper-Limb Motor Activities

The functions performed were adapted from prior research with chronic stroke patients [43]. The set of exercises included: 1. stacking blocks; 2. flipping scrapbook pages; 3. nine-hole pegboard; 4. grabbing saucepan and pouring water into a cup; and 5. opening hand to grasp and pick up a cup. The sequence of activities is summarized in Figure 3.

### 2.9. Side Effects following tDCS

A survey developed by Poreisz et al. [44] was used to examine side effects of tDCS (Table 1). Two hours after each tDCS session, the tDCS survey was given in the form of an interview. 

### 2.10. Statistical Analyses

The baseline demographic data between the control group and experimental group were compared for a statistically significant difference using the chi-squared test, paired samples *t*-test, and Student’s *t*-test. The data met the assumption of normality (Shapiro–Wilk test), Levene’s test of equality of variances, and homogeneity of regression slope, meaning we could conduct analysis of covariance (ANCOVA). ANCOVA was used to compare treatment effects between groups, with the baseline dataset as a covariate. The clinical significance of the effect of the intervention was calculated with the partial eta squared [45]. Using SPSS version 20.0, the data were analyzed (IBM Corp., Armonk, NY, USA). The level of alpha was set at 0.05.

## 3. Results

Sixty-four participants with chronic stroke participated in the study. All participants completed 10 treatment sessions for 2 weeks. Long-term follow-up could not be scheduled due to family commitments, work, and no-shows so assessments were merely conducted at baseline and after 2 weeks of treatment (Figure 1). At baseline (Table 2), there was no significant difference between the groups (control and experimental group) in demographic characteristics.

### 3.1. Within-Group Comparison (Paired Samples t-Test)

Within the experimental group, the FMA and ARAT scores showed significant improvements from baseline. The mean score of the FMA (upper limb) at baseline was 20.6 (SD = 2.6) and improved to 28.3 (SD = 6.9). Similarly, the mean score of ARAT at baseline was 43.00 (SD = 9.82) and improved to 52.20 (SD = 8.28). Within the control group, FMA and ARAT scores also showed significant improvements from baseline. The mean score of the FMA (upper limb) at baseline was 20.4 (SD = 3.7) and improved to 22.8 (SD = 5.0). Similarly, the mean score of ARAT at baseline was 17.4 (SD = 3.7) and improved to 18.9 (SD = 5.1). Table 3 is a summary of the statistics for the paired samples *t*-Test, the mean difference, and the confidence interval.

### 3.2. Between-Group Comparison (Analysis of Covariance)

A one-way ANCOVA was conducted to determine a statistically significant difference between the experimental and control groups in the FMA and ARAT scores, controlling for the baseline scores. There was also a significant effect of the dual intervention (MI and tDCS) on the post-test FMA scores after controlling for the effect of the baseline scores, F (414.4) = 35.79, *p* < 0.001; η^2^ = 0.37. Similarly, there was also a significant effect of combining the MI with the tDCS on the post-test ARAT scores after controlling for the effect of the baseline scores, F (440.09) = 37.46, *p* < 0.001; η^2^ = 0.38. The average change in scores of the FMA and the ARAT is summarized in Table 3.

### 3.3. Adverse Reaction following tDCS

Eight participants (six in the real-tDCS group and two in the sham-tDCS group) reported tingling sensations over the scalp. Five participants reported itching (five in the real-tDCS group) over the scalp. None of the participants reported any severe adverse events (Table 1).

## 4. Discussion

Based on the results of our study, anodal tDCS over the upper-limb motor cortex of the affected hemisphere, combined with MI and upper-limb functional training, yielded better results than the combination of MI and upper-limb functional training alone among patients with chronic stroke. Moreover, the treatment plan and procedure used to stimulate the cortical motor cortex were safe when applied to chronic stroke survivors. Yet, due to the limitations of our study design, we could not determine the unique contribution MI and/or upper-limb functional training made to the overall improvement of patients with stroke.

### 4.1. The Efficacy of Anodal tDCS in Conjunction with MI and Upper-Limb Functional Training

Studies report that a change in the FMA upper-limb scores of at least 5.2 is clinically significant [46]. In our study, the experimental group receiving tDCS with MI and upper-limb functional training achieved greater improvements than the clinically relevant thresholds on the FMA and ARAT scales (Figure 4). These results suggest that tDCS therapy, when combined with MI and upper-limb functional training, can significantly improve the upper-limb function in chronic stroke patients. tDCS is reported to be effective in a large number of quality studies with the hypothesis that combining peripheral and central inputs might promote neuroplasticity [47,48,49,50,51]. tDCS alone has demonstrated strong temporal excitability changes and motor gains [24]. Recent studies employing treatment paradigms using brain stimulation coupled with simultaneous peripheral stimulation have reported a wide variation in terms of functional gain. These variations in outcome variables could be due to variations in the treatment protocol such as in the electrodes, electrode size, intensity of the current, sample size, and location of cortical stimulation. 

Our study provides new evidence concerning the additive effect of combining therapies to decrease motor disability among stroke patients. Accordingly, it is beneficial for patients with chronic ischemic stroke to mentally rehearse an activity before performing it under tDCS. A study reported that tDCS combined with constraint-induced movement therapy led to an improvement in the FMA score by 6.3 points. This earlier study utilized a similar tDCS treatment procedure as the present study (30 min per day for 2 weeks) [42]. Compared to these previous reports, our study found a higher recovery in motor function on the affected side. This discrepancy could be due to variation in the tDCS protocol, the instrumentation, and the stage of stroke recovery. Our findings suggest combining anodal tDCS with MI and upper-limb functional training could improve motor function more than performing MI and upper-limb functional training alone. Future studies should be conducted including long–term follow-up to explore the retention of functional gain. 

It is worth noting that improvement in impairment (scores on FMA) was greater than the functional gain (scores on ARAT). These improvements in patients with chronic stroke were within a short span of 2 weeks, compared to similar research conducted among the stroke population with around 6 to 12 weeks’ duration [52]. The results of the present study suggest a temporal summative effect on the outcome measures of the ARAT and the FMA when tDCS is used concurrently with MI and upper-limb functional tasks. Similarly, a study conducted among 30 patients with stroke reported a timing-dependent interaction between tDCS and mirror therapy for upper-extremity motor recovery in patients with chronic stroke [53]. The authors reported that tDCS administered concurrently with mirror therapy is more beneficial than prior tDCS or sham tDCS [53]. A recent study conducted among 22 patients with chronic stroke receiving robotic therapy for three sessions per week for 6 weeks showed a 4.41-point change in FMA scores. This discrepancy could be due to the small sample size and post-stroke duration (458 days) [54]. Another study using a robotic rehabilitation module coupled with tDCS thrice per week for 6 weeks reported clinically significant improvement [55]. However, the procurement of expensive robotic equipment is not feasible in every rehabilitation clinic. The changes in the mean scores from baseline in the experimental group were 7.6 points (a 5.0% change in the mean score) for the FMA and 6.8 (a 3.8% change in the mean score) for the ARAT. These changes imply that tDCS with MI is more effective in reducing impairment than disability. Similar results have been reported for several combination therapies that combine tDCS with robotic therapy [56], constraint-induced movement therapy [57], nerve-muscle electrical stimulation [58], and feedback training [59].

### 4.2. Protocol

In our study, we used an intensity of 1.5 mA delivered by sponge electrodes over the affected primary motor area, which corresponds to the topographical representation of the upper limb. The tDCS stimulation lasted for 30 min, five times a week for two weeks. The procedure and current intensity were recommended by several researchers and are reported to deliver a current density of around 0.125 mA/cm^2^ [60], which is reported to be safe and sufficient to cause cortical modulation [61,62]. A study conducted among ten children with infantile stroke reported improvements in selective attention span after ten sessions of tDCS coupled with attention-improving activities. Furthermore, the authors reported that an intensity of 1.5 mA was safe for children with infantile stroke [63]. tDCS therapy modulates the cerebral cortex through two main cellular changes: the online mechanism, and the offline mechanism. The former changes the membrane potential, while the latter results in long-term depression at the anode.

### 4.3. Clinical Applicability

tDCS is a commonly used brain-stimulation method in clinics. Compared with transcranial magnetic stimulation, tDCS is more affordable, portable, and pragmatic. The tDCS protocol used in our study was safe and easy to apply. Although eight patients felt a tingling sensation and five experienced itching, no severe adverse effects were reported by the rest of the participants.

### 4.4. Limitations 

There were some drawbacks to this study. First, the study had a very small sample size. Second, our study was a double-blinded; however, the success of the blinding method was not evaluated, which may have introduced some biases into the outcomes. Third, it is possible that all patients received additional rehabilitation therapy outside of the clinical trial environment without the knowledge of investigators; however, none of the participants reported receiving additional treatment during the study period. Fourth, we accept that the design was not appropriate for discerning the discrete effect of MI and/or upper-limb functional training toward the overall improvement among patients with stroke. Future studies with more robust designs may help in understanding the discrete effects of MI and tDCS. Fifth, because both groups were made familiar with the tests at baseline, there was the potential for a practice effect in both groups.

## 5. Conclusions

Anodal tDCS stimulation over the affected primary motor cortex coupled with MI and upper-limb functional training reduces impairment and disability of the upper limbs among patients with chronic stroke.

## Figures and Tables

**Figure 1 ijerph-19-15199-f001:**
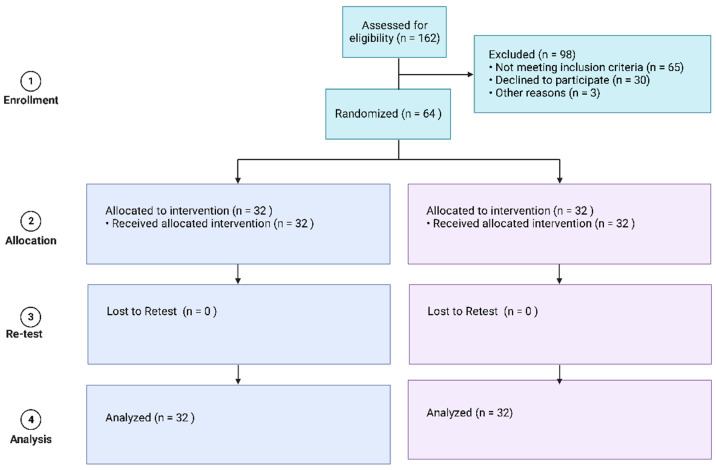
CONSORT diagram.

**Figure 2 ijerph-19-15199-f002:**
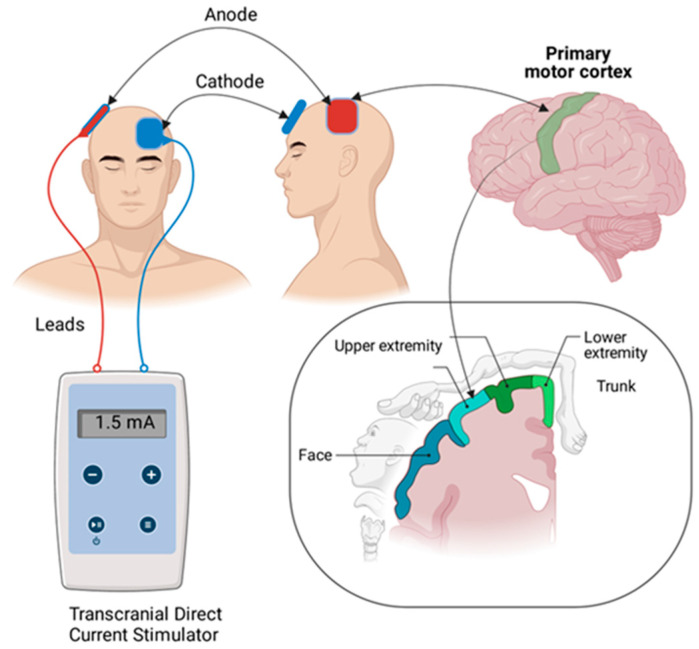
Placement of electrodes and corresponding cortical area of tDCS simulation.

**Figure 3 ijerph-19-15199-f003:**
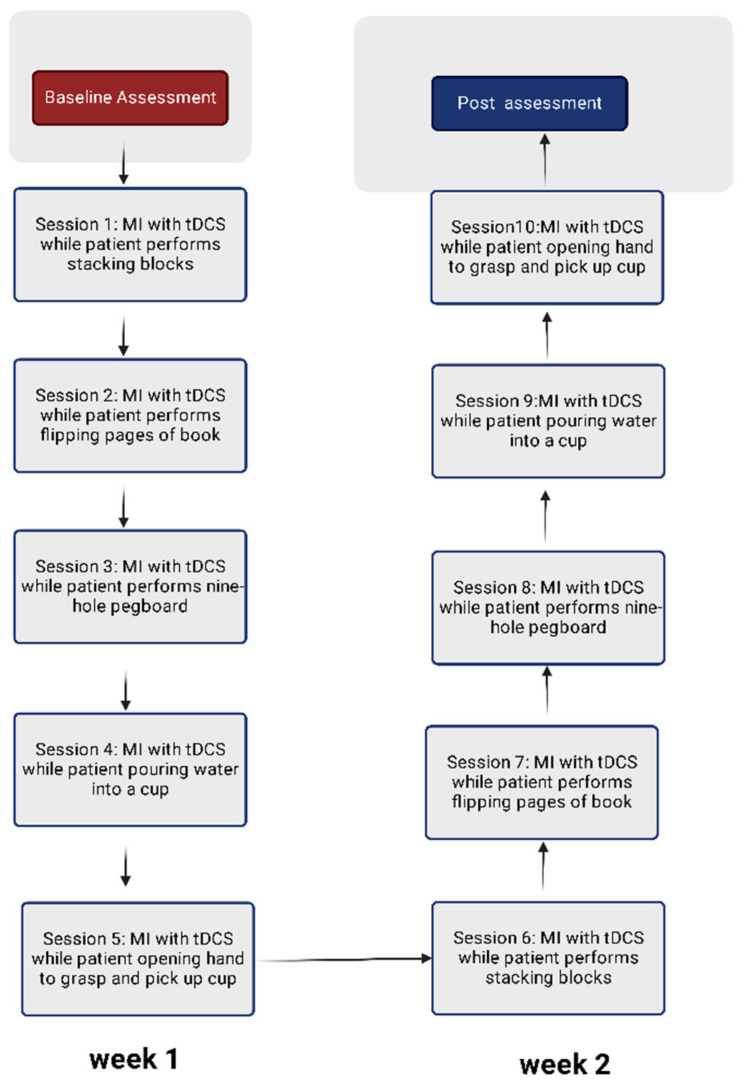
Regime of 10 sessions of treatment for two weeks.

**Figure 4 ijerph-19-15199-f004:**
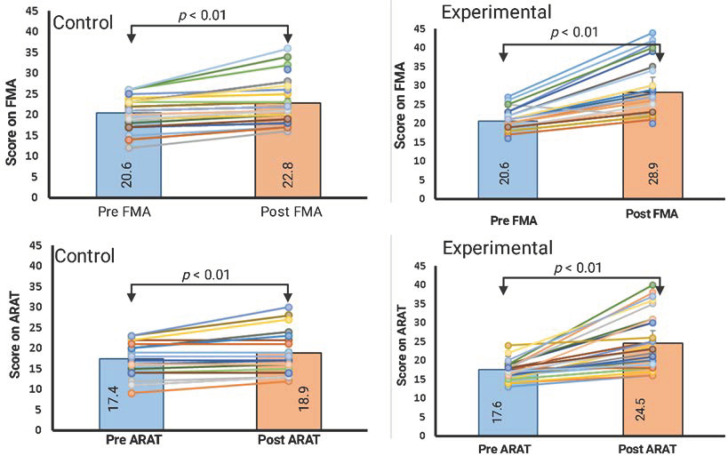
Pretest/post-test comparison of participants’ individual scores between experimental and control groups. ARAT; Action Research Arm Test, FMA; Fugl-Meyer Assessment.

**Table 1 ijerph-19-15199-t001:** Adverse reaction.

Adverse Effects	Number of Patients
Tingling sensation	8
Itching under electrodes	5
Fatigue following stimulation	0
Difficulties in concentrating	0
Nausea or vomiting	0
Insomnia	0
Burning sensation	0
Pain	0
Headache	0

**Table 2 ijerph-19-15199-t002:** Demographic characteristics of participants.

Variables	Control (*n* = 32)	Experimental (*n* = 32)	*p*
Age (years)	59.9 ± 5.6	58.7 ± 5.7	0.161 *
Gender (male/female)	24/8	25/7	0.500 ***
Mini-mental state examination	27.9 (0.89)	28.0 (0.8)	0.459 *
Onset (months)	7.4 ± 1.2	7.8 ± 1.3	0.145 *
Type of stroke			
Ischemic	30 (93.8)	28 (87.5)	0.306 ***
Hemorrhagic	2 (6.3)	4 (12.5)	
Side affected			
Right	12 (53.1)	18 (56.3)	
Left	15 (46.9)	14 (43.8)	
Associated risk factors			
Hypertension	21	18	0.069 **
Diabetes mellitus	4	4	
High cholesterol	0	5	
Cardiac disorders	5	2	
Others	2	3	
Region of stroke			
Cortical	1	3	0.613 ***
Subcortical	31	29	

Note: * chi-squared test; ** likelihood ratio; ND: not determined. *** Fisher’s exact test.

**Table 3 ijerph-19-15199-t003:** Between- and within-group comparisons.

Outcome	Experimental Group (*n* = 32)	Control Group (*n* = 32)	Between-Group Difference *p*-Value (η^2^ Value)
FMA score (upper limb)			
Baseline	20.6 (2.6)	20.4 (3.7)	*p* = 0.76 ^a^
Post-intervention	28.3 (6.9)	22.8 (5.0)	*p* < 0.001 (0.37) ^b^
Improvement	7.6	2.4	
95% confidence interval of the difference	5.9–9.4	1.6–3.3	
Standard error of mean	0.86	0.87	
Within-group difference *p*-value	*p* < 0.01 ^c^	*p* < 0.01 ^c^	
ARAT score			
Baseline	17.6 (2.6)	17.4 (3.7)	*p* = 0.78 ^a^
Post-intervention	24.5 (6.9)	18.9 (5.1)	*p* < 0.001 (0.38) ^b^
Improvement	6.8	1.5	
95% confidence interval of the difference	5.0–8.6	0.6–2.3	
Standard error of mean	0.42	0.41	
Within-group difference *p*-value	*p* < 0.01 ^c^	*p* < 0.01 ^c^	

FMA, Fugl-Meyer Assessment; ARAT, Action Research Arm Test; ^a^ between-group comparison at baseline by *t*-test; ^b^ between-group comparison of post intervention with analysis of covariance and effect size was assessed using the η^2^ value; ^c^ within-group comparison before and after intervention with paired samples *t*-test.

## Data Availability

The data presented in this study are available on request from the corresponding author.

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
