# Peer review of "Effect of Transcranial Direct Current Stimulation Augmented with Motor Imagery and Upper-Limb Functional Training for Upper-Limb Stroke Rehabilitation: A Prospective Randomized Controlled Trial"

_ijerph, 2022, doi:10.3390/ijerph192215199_

Round 1

Reviewer 1 Report

The study aimed to investigate the effects of motor imagery training for rehabilitation in conjunction with tdcs.

Moreover, the authors recruited a relatively large sample size of patients.

In line with the recent publication raising the possibility of applying tdcs on a supervised telerehabilitation treatment method online, Motor imagery training can have a promising affect on ensuring treatment in conjunction with tdcs. 

PITFALLS 

There are however some serious pitfalls with the design of the study.

There were two experimental groups, one with tdcs and motor imagery and the other with Sham tdcs and motor imagery. Both groups received motor imagery treatment. Therefore, based on these results, it is difficult to conclude whether the positive effects shown on the FMA scores and ART scores were direct improvements from the motor imagery training when the control group, who had no tdcs, also received motor imagery training. 

  I would recommend that the authors provide some background on choosing the design. 

The differences in improvement as seen in these resuls were more likely attributable to tdcs effects. 

The ideal method would have been to have a control group that received real tdcs in conjunction with conventional help extremity rehabilitation. The motor imagery will be in direct comparison with conventional upper extremity treatment. 

Instead of implying that motor imagery training is effective in promoting upper motor recovery in stroke patients, it would be more feasible to conclude that the Motor imagery training can be used as a useful adjunctive treatment modality during tdcs research studies. 

Also, extensive rewriting of the discussion is needed in order to prevent the Readers from a misunderstanding that this study demonstrates the effect of Motor images training per se when the results of the study indicate that tdcs was more effective when combined with motor imagery training than during sham tdcs.

 The discussion was too brief, and there seems to be more room to mention the limitations of this study. 

On the outcome parameters

Why were there only 2 outcome parameters? Compared to other upper extremity tdcs studies, the use two outcome parameters seem too brief and could be supported by more outcome parameters. 

Author Response

Response to Reviewer 1 Comments

Point 1: The study aimed to investigate the effects of motor imagery training for rehabilitation in conjunction with tdcs.

Moreover, the authors recruited a relatively large sample size of patients.

In line with the recent publication raising the possibility of applying tdcs on a supervised telerehabilitation treatment method online, Motor imagery training can have a promising affect on ensuring treatment in conjunction with tdcs.

Response 1: Thanks a lot for appreciating the importance of the study.

Point 2: PITFALLS 

There are however some serious pitfalls with the design of the study.

There were two experimental groups, one with tdcs and motor imagery and the other with Sham tdcs and motor imagery. Both groups received motor imagery treatment. Therefore, based on these results, it is difficult to conclude whether the positive effects shown on the FMA scores and ART scores were direct improvements from the motor imagery training when the control group, who had no tdcs, also received motor imagery training. 

  I would recommend that the authors provide some background on choosing the design. 

The differences in improvement as seen in these resuls were more likely attributable to tdcs effects

Response 2: We appreciate the reviewer's in-depth analysis and constructive comments on our article. We would like to elaborate on how our research design may help give an outcome that closely reflect our study's purpose. Our study's primary objective was to determine the efficacy of anodal tDCS when combined with MI and upper limb functional tasks as compared to the combination of MI and upper limb functional tasks alone among patients with chronic stroke. We can confidently attribute the improvement to the introduction of tDCS because both groups (control and experimental) performed MI and upper limb functional tasks. However, we accept that the design was not appropriate for discerning discreet effect of MI, and/or motor training on upper limb functional tasks towards the overall improvement among patients with stroke. Furthermore, we have added the limitation of the design in the limitation sections.   

Point 3:  The ideal method would have been to have a control group that received real tdcs in conjunction with conventional help extremity rehabilitation. The motor imagery will be in direct comparison with conventional upper extremity treatment.

Response 3: Thanks a lot for the comment, as per the purpose of our research study was to determine the effect of tDCS combined with MI and motor practice among patient with chronic stroke patients. Based on the results of our study we could most likely attribute the improvement in experimental group due to addition of tDCS as MI and motor practice is performed by both the groups. However, we could not determine the unique contribution of MI and motor practice of upper limb due to lack of comparative control group.

Point 4: Instead of implying that motor imagery training is effective in promoting upper motor recovery in stroke patients, it would be more feasible to conclude that the Motor imagery training can be used as a useful adjunctive treatment modality during tdcs research studies. 

Response 4: we completely agree with the reviewer, however the main focus of our study was to determine the efficacy of tDCS when combined with MI and upper limb functional tasks.

Point 5: Also, extensive rewriting of the discussion is needed in order to prevent the Readers from a misunderstanding that this study demonstrates the effect of Motor images training per se when the results of the study indicate that tdcs was more effective when combined with motor imagery training than during sham tdcs.

Response 5: Thanks a lot for pointing out, the sentences in the introduction and throughout the discussion is made clear about the purpose of the study. Conclusion is modified and in line with the results of our study.

Point 6: The discussion was too brief, and there seems to be more room to mention the limitations of this study.

Response 6: Thanks a lot for the comment, additional research and literature are added to the discussion section. and some important limitations added in the limitation section.

Point 7: On the outcome parameters

Why were there only 2 outcome parameters? Compared to other upper extremity tdcs studies, the use two outcome parameters seem too brief and could be supported by more outcome parameters. 

Response 7: Thanks a lot for the comment. We used only two comprehensive instruments for assessing upper limb impairment and disability because these instruments covered all the purposes of the research study.

Reviewer 2 Report

To the authors

General Comment

This paper focuses on the cumulative effects of combining motor imagery (MI) of upper limb functional activities with tDCS on chronic stroke patient’s upper limb rehabilitation. The authors found that tDCS combined with MI had a greater rehabilitation effect on several outcomes than just MI. The authors should be commended by the effort of recruiting, training and measuring 64 chronic stroke patients.

My main concern is related to the experimental design and the conclusions you can get from them. From the explanation in the methods section (lines 146-148) I understand that, in addition to tDCS (sham or anodal) and MI, all participants performed actual motor training. The authors conclude that MI and tDCS has a greater effect than just MI and that combining both rehabilitation tools could be beneficial. They can isolate the effects of tDCS with the inclusion of a Sham group. However, the effect of MI on the outcomes is unclear because all groups performed MI but also actual motor practice. Therefore I wonder how they can conclude that the effects are related to the combination of MI and tDCS and not just motor practice alone (control group) and motor practice with tDCS (experimental group). In order to conclude an additional effect of MI the experimental design should have included a group performing only motor practice (without MI). Therefore conclusions should be revisited or the actual rehabilitation protocol further explained. Some further comments are detailed below:

 INTRODUCTION

Introduction needs substantial changes. Specially the second and third paragraphs. You should improve the precision of the references you use (some of them has nothing to do with what you are saying, and there is a lack of citations of novel studies focus on the effects of tdcs on stroke patients motor function. See some specific comments below:

Line 59-60: As it is written, it seems that there are only two “clinically feasible treatments” (tdcs and MI) but it is not true (see for example strength-based cross-education rehabilitation interventions). So I suggest authors to reformulate their sentence.

 Lines 62-63: The reference that accompanies the sentence does not speaks about tDCS so it has to be changed.

 Line 63-65: You say that "much research" has evaluated the effects of tdcs on recovery after stroke but then you only cite a 2005 study.

 You should include more recent studies, including several meta-analysis focused on the effects of tdcs over motor function in stroke patients. Here are some examples:

 Bastani A, Jaberzadeh S (2012) Does anodal transcranial direct current stimulation enhance excitability of the motor cortex and motor function in healthy individuals and subjects with stroke: A systematic review and meta-analysis. Clin Neurophysiol 123:644–657. https://doi.org/10.1016/j.clinph.2011.08.029

 Dong K, Meng S, Guo Z, et al (2021) The Effects of Transcranial Direct Current Stimulation on Balance and Gait in Stroke Patients: A Systematic Review and Meta-Analysis   . Front. Neurol.   12

 Li Y, Fan J, Yang J, et al (2018) Effects of transcranial direct current stimulation on walking ability after  stroke: A systematic review and meta-analysis. Restor Neurol Neurosci 36:59–71. https://doi.org/10.3233/RNN-170770

 Van Hoornweder S, Vanderzande L, Bloemers E, et al (2021) The effects of transcranial direct current stimulation on upper-limb function  post-stroke: A meta-analysis of multiple-session studies. Clin Neurophysiol  Off J Int Fed  Clin Neurophysiol 132:1897–1918. https://doi.org/10.1016/j.clinph.2021.05.015

Lines 66-67: This sentence is repeated above. Delete the first one. I suggest rewritting to focus first on explaining Motor imagery (second paragraph), then, in the next paragraph, explain tdcs and introduce the potential of combining the two methods. That would made easier to follow the logic of your hypothesis.

 Lines 75-77: Those references does not focus on the effects of tdcs on stroke patients. I suggest review them.

 Lines 81-83: This sentence is not understandable in the context of the paragraph. What do you mean with "interhemispheric inhibition". You should further explain the concept.

METHODS

Specific comments:

Did the patients do a familiarization session with the tests? If not, part of the increase in the scores could be related just to familiarization with the tests

Line 120: I guess this sould be 0.05.

Lines 146-148: This should be further clarified. Did the patients performed actual motor training? If so, conclusions about the effect of MI could be flawed. See the general comments above.

Figure 3: Usually figures are readed from left to right. Time lapse is also usually represented from left to right. I suggest then to modify the figure in order to read it from left to right.

RESULTS

Figure 2: I strongly suggest to reconsider modifications to this figure. No standard deviations are shown, just the mean (in numbers and in the bars), an information that is already shown in Table 3 (so this figure ads no extra value). Graphics showing individual scores are preferred to just mean bars. I strongly suggest to show individual responses (e.g. dots) together with the mean responses (e.g bars). You can also consider to plot FMA PRE-POST and ARAT PRE-POST for each group and linking with lines each individual dot to see the tendency of change of each individual. Those kind of figures are far more useful than just show the mean, which indeed is redundant with Table 3 information.

DISCUSSION

Lines 240-241: The reference cited does not seem to support the four point minimal change to be considered as clinically significant. Could you please provide the exact reference to support this statement? There are, for example, studies showing a minimal of 12 points to be perceived as meaningful in stroke patients. Reference:

Hiragami S, Inoue Y, Harada K (2019) Minimal clinically important difference for the Fugl-Meyer assessment of the  upper extremity in convalescent stroke patients with moderate to severe hemiparesis. J Phys Ther Sci 31:917–921. https://doi.org/10.1589/jpts.31.917

Lines 243-244, 262-262: These statements should be revisited according to the main general comment (see above).

Lines 301-302: In the methods section you say that physiotherapists were blinded to tDCS condition. Then here you say that the single-blind (only patients?) could have influenced the outcomes. Please clarify blinding procedures and, if physiotherapists were blinded, how can this introduce bias in the outcomes?

Lines 302-303: You can not forbid patients to get additional rehabilitation outside of the study, however you may should had to record through questionnaires if they received additional treatments and report them.

Author Response

Response to Reviewer 2 Comments

Point 1: This paper focuses on the cumulative effects of combining motor imagery (MI) of upper limb functional activities with tDCS on chronic stroke patient’s upper limb rehabilitation. The authors found that tDCS combined with MI had a greater rehabilitation effect on several outcomes than just MI. The authors should be commended by the effort of recruiting, training and measuring 64 chronic stroke patients.

Response 1: Thanks a lot for your appreciation.

Point 2: My main concern is related to the experimental design and the conclusions you can get from them. From the explanation in the methods section (lines 146-148) I understand that, in addition to tDCS (sham or anodal) and MI, all participants performed actual motor training. The authors conclude that MI and tDCS has a greater effect than just MI and that combining both rehabilitation tools could be beneficial. They can isolate the effects of tDCS with the inclusion of a Sham group. However, the effect of MI on the outcomes is unclear because all groups performed MI but also actual motor practice. Therefore, I wonder how they can conclude that the effects are related to the combination of MI and tDCS and not just motor practice alone (control group) and motor practice with tDCS (experimental group).

Response 2: We appreciate the reviewer's in-depth analysis and constructive comments on our article. We would like to elaborate on how our research design may help give an outcome that closely reflect our study's purpose. Our study's primary objective was to determine the efficacy of anodal tDCS when combined with MI and upper limb functional tasks as compared to the combination of MI and upper limb functional tasks alone among patients with chronic stroke. We would attribute the improvement to the introduction of tDCS because both groups (control and experimental) performed MI and upper limb functional tasks. However, we accept that the design was not appropriate for discerning discreet effect of MI, and/or motor training on upper limb functional tasks towards the overall improvement among patients with stroke. Furthermore, we have added the limitation of the design in the limitation sections.   

Point 3: In order to conclude an additional effect of MI the experimental design should have included a group performing only motor practice (without MI). Therefore, conclusions should be revisited or the actual rehabilitation protocol further explained. Some further comments are detailed below:

Response 3: Agree, it would have been desirable to include a third group that received only upper limb motor training. However, we only needed two groups to address the research question of our study, which was to determine the efficacy of anodal tDCS when combined with MI and upper limb functional tasks as compared to MI and upper limb functional training.

We fully agree with the reviewer that the study's conclusion may be misleading. Consequently, we revised the conclusion. In addition, However, we accept that the design was not suitable for discerning discreet effect of MI, and/or motor training on upper limb functional training.

Point 4: Introduction needs substantial changes. Specially the second and third paragraphs. You should improve the precision of the references you use (some of them has nothing to do with what you are saying, and there is a lack of citations of novel studies focus on the effects of tdcs on stroke patients motor function. See some specific comments below:

Response 4: The changes are made in the manuscript as recommended.

Point 5: Line 59-60: As it is written, it seems that there are only two “clinically feasible treatments” (tdcs and MI) but it is not true (see for example strength-based cross-education rehabilitation interventions). So I suggest authors to reformulate their sentence.

Response 5: The changes are made in the manuscript as recommended.

Point 6: Lines 62-63: The reference that accompanies the sentence does not speaks about tDCS so it has to be changed.

Response 6: The changes are made in the manuscript as recommended.

Point 7: Line 63-65: You say that "much research" has evaluated the effects of tdcs on recovery after stroke but then you only cite a 2005 study.

 You should include more recent studies, including several meta-analysis focused on the effects of tdcs over motor function in stroke patients. Here are some examples:

 Bastani A, Jaberzadeh S (2012) Does anodal transcranial direct current stimulation enhance excitability of the motor cortex and motor function in healthy individuals and subjects with stroke: A systematic review and meta-analysis. Clin Neurophysiol 123:644–657. https://doi.org/10.1016/j.clinph.2011.08.029

 Dong K, Meng S, Guo Z, et al (2021) The Effects of Transcranial Direct Current Stimulation on Balance and Gait in Stroke Patients: A Systematic Review and Meta-Analysis   . Front. Neurol.   12

 Li Y, Fan J, Yang J, et al (2018) Effects of transcranial direct current stimulation on walking ability after  stroke: A systematic review and meta-analysis. Restor Neurol Neurosci 36:59–71. https://doi.org/10.3233/RNN-170770

 Van Hoornweder S, Vanderzande L, Bloemers E, et al (2021) The effects of transcranial direct current stimulation on upper-limb function  post-stroke: A meta-analysis of multiple-session studies. Clin Neurophysiol  Off J Int Fed  Clin Neurophysiol 132:1897–1918. https://doi.org/10.1016/j.clinph.2021.05.015

Response 6: The references suggested included in the article.

Point 7: Lines 66-67: This sentence is repeated above. Delete the first one. I suggest rewritting to focus first on explaining Motor imagery (second paragraph), then, in the next paragraph, explain tdcs and introduce the potential of combining the two methods. That would made easier to follow the logic of your hypothesis.

Response 7: The changes are made in the manuscript as recommended.

Point 8: Lines 75-77: Those references does not focus on the effects of tdcs on stroke patients. I suggest review them.

Response 8: The relevant references cited.

Point 9: Lines 81-83: This sentence is not understandable in the context of the paragraph. What do you mean with "interhemispheric inhibition". You should further explain the concept.

Response 9: Thanks a lot for pointing that out. Interhemispheric inhibition refers to the neurophysiological mechanism in which one hemisphere of the brain inhibits the opposite hemisphere. The sentence was not relevant to the context of the paragraph; therefore, it was removed.

Point 10: METHODS

Specific comments:

Did the patients do a familiarization session with the tests? If not, part of the increase in the scores could be related just to familiarization with the tests

Response 10: Thanks a lot for the comment, we agree with the reviewers that practice tests would result in an increase in the scores, but both groups were provided with an equal practice session. Moreover, there was no repeated assessment of patients, which might have led to considerable improvement in scores. The pre-assessment with familiarization session was conducted at baseline and at the end of the 10th session. However, a little improvement might be of some significance, so we added a sentence in the limitation section about the possible practice effect or learning effect among both the groups.

Point 11: Line 120: I guess this sould be 0.05.

Response 11: Thanks, corrected as recommended

Point 12: Lines 146-148: This should be further clarified. Did the patients performed actual motor training? If so, conclusions about the effect of MI could be flawed. See the general comments above.

Response 12: Yes, both groups performed MI and upper limb functional training. we accept that the design was not appropriate for discerning discreet effect of MI, and/or motor training on upper limb functional tasks towards the overall improvement among patients with stroke. Furthermore, we have added the limitation of the design in the limitation sections.

Point 13: Figure 3: Usually figures are readed from left to right. Time lapse is also usually represented from left to right. I suggest then to modify the figure in order to read it from left to right.

Response 13: Thanks, the figure modified as recommended

Point 14: Figure 2: I strongly suggest to reconsider modifications to this figure. No standard deviations are shown, just the mean (in numbers and in the bars), an information that is already shown in Table 3 (so this figure ads no extra value). Graphics showing individual scores are preferred to just mean bars. I strongly suggest to show individual responses (e.g. dots) together with the mean responses (e.g bars). You can also consider to plot FMA PRE-POST and ARAT PRE-POST for each group and linking with lines each individual dot to see the tendency of change of each individual. Those kind of figures are far more useful than just show the mean, which indeed is redundant with Table 3 information

Response 14: We agree with the reviewers about the figure 2 being redundant. We have created a new graph with dots and mean bar that represented the individual scores of the participants.

Point 15: Lines 240-241: The reference cited does not seem to support the four point minimal change to be considered as clinically significant. Could you please provide the exact reference to support this statement? There are, for example, studies showing a minimal of 12 points to be perceived as meaningful in stroke patients. Reference:

Hiragami S, Inoue Y, Harada K (2019) Minimal clinically important difference for the Fugl-Meyer assessment of the  upper extremity in convalescent stroke patients with moderate to severe hemiparesis. J Phys Ther Sci 31:917–921. https://doi.org/10.1589/jpts.31.917

Response 15: Thanks a lot for the reference; however, the recommended reference reporting the 12.4 points as MCID was for acute stroke patients. But our participants had chronic stroke; therefore, we have cited a research study involving patients with chronic stroke who reported 5.2 points (FMA) as MCID [1]. The amendments are made as per the reference.

Reference

Page SJ, Fulk GD, Boyne P. Clinically important differences for the upper-extremity Fugl-Meyer Scale in people with minimal to moderate impairment due to chronic stroke. Phys Ther. 2012;92(6):791–8.

Point 16: Lines 243-244, 262-262: These statements should be revisited according to the main general comment (see above).

Response 16: The sentence in the manuscript says, "When anodal tDCS was combined with MI and upper limb functional training, patients with chronic stroke showed significant improvements in impairment and function compared to when they were treated with MI and upper limb functional training alone." 

Point 17: Lines 301-302: In the methods section you say that physiotherapists were blinded to tDCS condition. Then here you say that the single-blind (only patients?) could have influenced the outcomes. Please clarify blinding procedures and, if physiotherapists were blinded, how can this introduce bias in the outcomes?

Response 17: Thanks a lot for pointing out, the sentence is corrected as double blinded. Both the physiotherapist and the patients were unaware of the allocation. However, we were unable to determine the success of our blinding procedure, which may have introduced bias.

Point 18: Lines 302-303: You can not forbid patients to get additional rehabilitation outside of the study, however you may should had to record through questionnaires if they received additional treatments and report them.

Response 18: Thanks a lot for the comment, our study period was brief (10 sessions for 2 weeks). Upon signing the consent form, participants were requested not to undergo any additional treatment for their upper limb for the next two weeks. According to participant-investigator communication, none of our participants received additional treatment for their upper limbs outside of the study. It might be possible that a participant received intervention without the investigator's knowledge.

Round 2

Author Response

Response to Reviewer 2 Comments

Point 1: I think the reorganized intro is easier to understand. However I suggest to better link some segments. For example: (this recommendations are just suggestions, authors should feel free to introduce them as they are wrote below or modify them as they consider)

Response 1: Thanks a lot for the suggestions, all changes recommended by reviewers are implemented in the manuscript.

Point 2: Lines 73-76: “A systematic review of four randomized controlled trials and one clinical trial reported a positive effect of mental rehearsal on arm function after stroke [16], notwithstanding coupling MI with other treatments may be more effective than motor imagery alone [17]

Response 2: Changed as recommended

Point 3: Line 77: “One of those treatments that could be easily coupled with MI is transcranial direct current stimulation (tDCS). tDCS is a noninvasive…”

Response 3: Changed as recommended

Point 4: Line 80: “The study conducted For example, one study using tDCS reported…”

Response 4: The changes are made in the manuscript as recommended.

Point 5: Line 88: “Therefore, MI paired with…”

Response 5: The changes are made in the manuscript as recommended.

 Point 6: Lines 92-95: I suggest to combine both sentences and include a hypothesis

Response 6: The sentence combined and a hypothesis included in the manuscript.

Point 7: METHODS

Line 148: Should not the CONSORT Diagram be the Figure 1?

Response 6: Changed as recommended.

Point 7: RESULTS

Figure 3: I was thinking more about this kind of graph (see picture):

Response 7: A new figure is made as recommended.

Point 8: DISCUSSION

LINE 297: “…the unique contribution of MI and/or upper limb…”

Response 8: Changed as recommended.

Point 9: LINE 301: In your response to my previous comments you mention 5.2 points, being the 12.4 the MCID for acute stroke patients.

Response 9: Thanks a lot for pointing out. The 12.4 MCID changed to 5.2

Point 10: Title

Authors may reconsider the title due to motor imagery was combined also with functional

upper limb training so they can not conclude if the benefits of the control and the

experimental group were related to motor imagery alone (both did actual motor practice)

Response 10: Thanks a lot for the comment, we included the upper limb training in the title as “

Effect of Transcranial Direct Current Stimulation Augmented with Motor Imagery and Upper Limb Functional training for Upper Limb Stroke Rehabilitation: A Prospective Randomized Controlled Trial”
